# SIMPLIFYING GNN PERFORMANCE WITH LOW RANK KERNEL MODELS

## ABSTRACT

We revisit recent spectral GNN approaches to semi-supervised node classification (SSNC). We posit that many of the current GNN architectures may be over-engineered. Instead, simpler, traditional methods from nonparametric estimation, applied in the spectral domain, could replace many deep-learning inspired GNN designs. These conventional techniques appear to be well suited for a variety of graph types reaching state-of-the-art performance on many of the common SSNC benchmarks. Additionally, we show that recent performance improvements in GNN approaches may be partially attributed to shifts in evaluation conventions. Lastly, an ablative study is conducted on the various hyperparameters associated with GNN spectral filtering techniques.

## 1 INTRODUCTION

The problem of semi-supervised node classification (SSNC) (Seeger, 2002; Belkin et al., 2006) has been a focal point in graph-based semi-supervised learning. Modern approaches to node classification on graphs make use of complex Graph Neural Networks (GNNs) (Scarselli et al., 2009) for prediction. These networks are trained to predict node labels, drawing on both the individual features of nodes and the broader network structure. From a statistical standpoint, SSNC represents a compelling regression or classification problem that incorporates network information.

The fundamental premise of SSNC is that the network structure ($A$) allows us to borrow information from the neighbors of nodes for which we lack a response. This borrowing can enhance the prediction of the unobserved responses beyond what could be achieved with a traditional regression of $y_i$ on $x_i$. Recently, there has been a wide breadth of literature (Veličković et al., 2018; Chien et al., 2021; Luan et al., 2022) which attempt to leverage network structure using GNNs. This recent flurry of activity has led to the proposal of many competing, often complex, architectures to solve the SSNC problem.

In this paper, we review top-of-the-leaderboard, benchmarking practices and confirm whether or not this "zoo" of models is necessary to achieve SOTA-like results. Recent studies by Maurya et al. (2022) and Wang & Zhang (2022) have suggested that simple spectral approaches may be sufficient to achieve SOTA performance for semi-surpervised graph classification. Using standard techniques from functional estimation, we simultaneously simplify and generalize previous spectral approaches to SSNC while maintaining or exceeding previous performance benchmarks. In particular, we are able to achieve improvements of +5% and +20% compared to other spectral methods on directed networks such as Chameleon and Squirrel (Rozemberczki et al., 2021).

Our contributions are as follows:

- Highlight spectral reshaping and modeling techniques which generalize previous spectral filtering approaches.
- Outline common evaluation practices which have an outsized effect on model performance.
- Simplify modeling hyperparameters (e.g. dropout probabilities, model depth, parameter-specific optimizers) while retaining SOTA or near-SOTA performance.

By standardizing evaluation practices and simplifying modeling considerations, we aim to disambiguate performance in the GNN model-space and believe our results will lead to more interpretable models and heuristics for future SSNC problems.

## 2 GNN AND SSNC FORMALISM

Consider a network represented by an adjacency matrix $A$ on $n$ nodes. Each node $i$ in the network is associated with a feature vector $\boldsymbol{x}_i \in \mathbb{R}^d$ and a response $y_i \in \mathbb{R}$. The collection of $n$ feature vectors will be succinctly expressed as $\boldsymbol{X} = [\boldsymbol{x}_1, \ldots, \boldsymbol{x}_n]^T \in \mathbb{R}^{n \times d}$. In the case of SSNC, all node features $\boldsymbol{X}$, the observed network $\boldsymbol{A}$, and a subset of the responses $(y_i)_{i \in S}$ are known. The goal of SSNC will be to correctly predict unobserved responses $(y_i)_{i \in S^c}$ given the previously stated knowns.

A mainstay of all GNN architectures is the feature propagation structure $\boldsymbol{P} \in \mathbb{R}^{n \times n}$. Common choices of $\boldsymbol{P}$ include the adjacency matrix $\boldsymbol{A}$ and its transformed variants, e.g. normalized Laplacian. These propagation structures need not be static. Indeed there are popular GNN architectures (Veličković et al., 2018) which introduce layer-dependent interactions between a base propagation $\boldsymbol{P}^0$ and intermediate features $\boldsymbol{Z} \in \mathbb{R}^{n \times d'}$.

If we abstract away the aggregation specifics of propagations $\{\boldsymbol{P}^\ell\}_\ell$, then intermediate representations of most GNNs can be recursively expressed as

$$\boldsymbol{Z}^{\ell+1} = \phi \odot (\boldsymbol{P}^\ell \boldsymbol{Z}^\ell \boldsymbol{W}^\ell) \qquad \text{for layers } \ell = 1, \ldots, L, \tag{1}$$

where $\boldsymbol{W}^\ell \in \mathbb{R}^{d_\ell \times d_{\ell+1}}$ are weight matrices and $\phi : \mathbb{R} \to \mathbb{R}$ is a scalar function which is to be applied element-wise. In the case of a $C$-class classification, it is common to extract row-wise "argmax"'s of the final features $\boldsymbol{Z}^L \in \mathbb{R}^{n \times C}$ using differentiable argmax surrogates such as $\mathrm{softmax}$.

Our studies will consider the simplest variant of GNN: a one layer, linear GNN, that is $\phi = \mathrm{id}$, where special attention is paid to the propagation structure $\boldsymbol{P}$. We will consider fixed and learnable propagation structures derived from variants of the adjacency matrix $\boldsymbol{A}$. Throughout, we will make use of spectral and singular value decompositions (SVD) where, in the case of SVD, $\boldsymbol{A} = \boldsymbol{U} \boldsymbol{\Sigma} \boldsymbol{V}^T$ with $\boldsymbol{\Sigma} = \mathrm{diag}(\sigma_i)$ and $\sigma_1 \geq \cdots \geq \sigma_n$ are the singular values of $\boldsymbol{A}$. In our analysis, we will consider combinations of low-rank

$$\boldsymbol{A}^{(r)} = \boldsymbol{U}_{:r} \boldsymbol{\Sigma}_{:r} (\boldsymbol{V}_{:r})^T$$

and kernelized

$$\boldsymbol{P}^{(\mathcal{K})} = \boldsymbol{U}(\mathrm{diag}(\boldsymbol{K}\boldsymbol{\alpha}))\boldsymbol{V}^T$$

representations of the network $\boldsymbol{A}$. In the kernelized case, $\boldsymbol{\alpha} \in \mathbb{R}^n$ is a trainable free parameter and $K_{i,j} = \mathcal{K}(\sigma_i, \sigma_j)$ is a kernel matrix formed by applying a kernel function $\mathcal{K} : \mathbb{R} \times \mathbb{R} \to \mathbb{R}$ to the singular values of $A$. The idea is that we can achieve a reshaping of the spectrum $h(\sigma_i)$ by a general function $h$ through an appropriate choice of the kernel function and $\boldsymbol{\alpha}$ such that $h(\sigma_i) = (\boldsymbol{K}\boldsymbol{\alpha})_i$. This is motivated by the so-called representer theorem (Schölkopf et al., 2001) which holds valid if $h$ belongs to reproducing kernel Hilbert space of continuous functions $\mathbb{H}$.

### 2.1 MOTIVATING SPECTRAL METHODS AND LEARNABLE PROPAGATIONS

Implicit in all graph learning problems, is the assumption that the nodal features $\boldsymbol{X}$ are only partially informative towards learning the response $\boldsymbol{y}$. Regression on the full set of observations $(\boldsymbol{X}, \boldsymbol{A})$ is expected to lead to better response outcomes, but without knowledge of the underlying graph generation process it becomes difficult to determine how observation $\boldsymbol{A}$ should be included in our modeling. Nevertheless, there are some broad strokes we can make when talking about generation of network data. Consider the following, similar lenses from which we can view network data generation:

- The feature-network pair $(\boldsymbol{X}, \boldsymbol{A})$ share a dependent structure. That is, we may assume there is a correlation between pairs of features $(\boldsymbol{x}_i, \boldsymbol{x}_j)$ and the appearance of corresponding edges $A_{i,j}$.

- The spectral representation of $\boldsymbol{A}$ can be used to form meaningful partitions on the set of nodes $[n]$. These partitions may vary depending on the graph learning task at hand.

The first view is natural and leads to considering propagation schemes between features $\boldsymbol{X}$ and a propagation $\boldsymbol{P}$ formed from polynomial and algebraic combinations of the the adjacency matrix $\boldsymbol{A}$. The second view point is primarily motivated by analysis done in community detection, where class clusters of certain processes like the stochastic block model (SBM) (Holland et al., 1983) and

| Dataset | Cora | Citeseer | Pubmed | Chameleon | Squirrel | Actor | Cornell | Texas | Wisconsin |
|---|---|---|---|---|---|---|---|---|---|
| Nodes | 2708 | 3327 | 19717 | 2277 | 5201 | 7600 | 183 | 183 | 251 |
| Edges | 5429 | 4732 | 44338 | 36101 | 217073 | 33544 | 295 | 309 | 499 |
| Features | 1433 | 3703 | 500 | 2325 | 2089 | 931 | 1703 | 1703 | 1703 |
| Classes | 7 | 6 | 3 | 5 | 5 | 5 | 5 | 5 | 5 |

Table 1: Summary statistics on benchmark networks, provided by Pei et al. (2020).

the random dot product graphs (RDPG) Young & Scheinerman (2007) can be determined using the spectral information of the observed graph. This second view point leads to the consideration of *graph Fourier* methods (Shuman et al., 2013; Ricaud et al., 2019) using the spectral data found in $\boldsymbol{A}$. Of the two approaches, the graph Fourier methods are more general and will be our focus when constructing learnable propagation stuctures $\boldsymbol{P}$.

In graph Fourier methods, considering the undirected case for simplicity, a learnable filter $h : \mathbb{R} \to \mathbb{R}$ is applied to the eigenvalue-eigenvector pairs $(\lambda_i, \boldsymbol{u}_i)$ of $\boldsymbol{A}$ to construct the propagation operator

$$\boldsymbol{P} = \sum_{i=1}^{n} h(\lambda_i) \boldsymbol{u}_i \boldsymbol{u}_i^T. \tag{2}$$

A sufficiently general and practical family of functions to estimate $h$ from could be a reproducing kernel Hilbert space (RKHS) of continuous functions $\mathbb{H}$ with associated kernel function $\mathcal{K}$. The choices of kernel $\mathcal{K}$ is flexible and determines the kind of regularity we wish to impose on graph the spectral domain. Important to note however, is that our point evaluations $h(\lambda_i) = (\boldsymbol{K}\boldsymbol{\alpha})_i$ are dependent on a kernel matrix $K$ which is created using noisy observations $(\lambda_i)_i$. For this reason we will also consider truncations $r \leq n$ of the form

$$\boldsymbol{P}^{(r,\mathcal{K})} = \sum_{i=1}^{r} (\boldsymbol{K}\boldsymbol{\alpha})_i \boldsymbol{u}_i \boldsymbol{u}_i^T, \tag{3}$$

with eigenvalue-eigenvector pairs $(\lambda_i, \boldsymbol{u}_i)$ being ordered according to their eigenvalue magnitude in decreasing order.

## 3 EXPERIMENTS

Our modeling efforts will be specific to the propagation structure $\boldsymbol{P}$ with no modifications made on the original features $X$ or linear weights $\boldsymbol{W}$. In our experiments we do not consider any model augmentations such as dropout (Srivastava et al., 2014), batchnorm (Ioffe & Szegedy, 2015), or per-parameter optimizers (i.e. different learning rates for different layers). The design of $\boldsymbol{P}$ will have the following degrees of freedom:

- **Matrix representation of network**. We will consider adjacency $\boldsymbol{A}$ and Laplacian $\boldsymbol{D} - \boldsymbol{A}$ and their normalized variants. Here $\boldsymbol{D}$ is column-wise sums of $\boldsymbol{A}$ placed in diagonal matrix format.

- **Spectral truncation factor**. Given a truncation factor $r$, the spectral system $(\boldsymbol{U}, \boldsymbol{\Lambda})$, resp. $(\boldsymbol{U}, \boldsymbol{\Sigma}, \boldsymbol{V}^T)$, will be reduced to $(\boldsymbol{U}_{:r}, \boldsymbol{\Lambda}_{:r})$, resp. $(\boldsymbol{U}_{:r}, \boldsymbol{\Sigma}_{:r}, (\boldsymbol{V}_{:r})^T)$, where the eigenvectors associated with the bottom $n - r$ eigenvalue magnitudes are dropped. In our experiments, truncation factors from 0 to 95% in 5% intervals will be considered.

- **Choice of kernel**. Some kernels we will consider are the identity ($\mathbf{1}\{i = j\}$), linear ($\lambda_i \lambda_j$), compact Sobolev ($\min(\lambda_i, \lambda_j)$), unbounded Sobolev ($\exp(\gamma|\lambda_i - \lambda_j|)$), and Gaussian radial basis function (RBF) ($\exp(\gamma|\lambda_i - \lambda_j|^2)$) kernels. Note that the identity kernel does not generate a continuous RKHS. In the case of the last two kernels, bandwidth parameter $\gamma \in \mathbb{R}_+$ will be selected using a validation process.

Our methods are evaluated against common SSNC benchmark datasets, summarized in Table 1.More information on the benchmarks can be found in Pei et al. (2020). All values are recorded using the *balanced splits* defined in Chien et al. (2021). Section 5 further provides a comprehensive analysis on the impact of the splitting conventions.

| Directed Networks | | | |
| --- | --- | --- | --- |
| | Chameleon | Squirrel | Actor |
| MLP2 | $48.5 \pm 2.6$ | $34.8 \pm 1.4$ | $40.3 \pm 2.3$ |
| LINEAR | $48.1 \pm 3.2$ | $34.9 \pm 1.4$ | $38.9 \pm 1.2$ |
| PROP. LINEAR | $79.0 \pm 1.4$ | $78.0 \pm 1.1$ | $32.4 \pm 1.3$ |
| KERNEL | $78.7 \pm 1.1$ | $76.0 \pm 1.2$ | $32.2 \pm 1.8$ |
| LR KERNEL | $79.4 \pm 1.4$ | $76.8 \pm 1.3$ | $32.3 \pm 1.7$ |
| GPRGNN* | $67.5 \pm 0.4$ | $49.9 \pm 0.5$ | $39.3 \pm 0.3$ |
| JACOBICONV* | $74.2 \pm 1.0$ | $55.8 \pm 0.6$ | $40.7 \pm 1.0$ |
| ACMII-GCN | $68.4 \pm 1.4$ | $54.5 \pm 2.1$ | $41.8 \pm 1.2$ |

| Undirected Networks | | | | | | |
| --- | --- | --- | --- | --- | --- | --- |
| | Cora | CiteSeer | PubMed | Cornell | Texas | Wisconsin |
| MLP2 | $77.8 \pm 1.6$ | $77.2 \pm 1.1$ | $88.2 \pm 0.5$ | $86.1 \pm 3.0$ | $91.7 \pm 4.4$ | $95.0 \pm 2.6$ |
| LINEAR | $78.9 \pm 2.0$ | $76.2 \pm 1.2$ | $85.8 \pm 0.4$ | $84.9 \pm 5.6$ | $89.7 \pm 3.8$ | $95.0 \pm 3.8$ |
| PROP. LINEAR | $84.0 \pm 2.0$ | $73.9 \pm 1.4$ | $82.6 \pm 0.5$ | $67.8 \pm 8.7$ | $86.8 \pm 3.5$ | $83.8 \pm 3.2$ |
| KERNEL | $88.6 \pm 1.0$ | $81.1 \pm 1.0$ | $89.4 \pm 0.8$ | $83.3 \pm 5.9$ | $88.2 \pm 2.6$ | $92.1 \pm 3.4$ |
| LR KERNEL | — | — | — | — | — | — |
| GPRGNN* | $79.5 \pm 0.4$ | $67.6 \pm 0.4$ | $85.1 \pm 0.1$ | $91.4 \pm 0.7$ | $92.9 \pm 0.6$ | NA |
| JACOBICONV* | $89.0 \pm 0.5$ | $80.8 \pm 0.8$ | $89.6 \pm 0.4$ | $92.3 \pm 2.8$ | $92.8 \pm 2.0$ | NA |
| ACMII-GCN | $89.0 \pm 0.7$ | $81.8 \pm 1.0$ | $90.7 \pm 0.5$ | $95.9 \pm 1.8$ | $95.1 \pm 2.0$ | $96.6 \pm 2.4$ |

Table 2: Performance: Mean test accuracy $\pm$ std. dev. over 10 data splits. Models include our own variations of "Linear" and "Propagated Linear" GNNs, along with other state-of-the-art (SOTA) GNNs. Dashed entry in for LR KERNEL signifies validated choice is the same as the full-rank KERNEL. Performance is comparable between our simple GNNs and SOTA in some cases. Results for GPRGNN, JACOBICONV and ACMII-GCN are cited from Chien et al. (2021), Wang & Zhang (2022), and Luan et al. (2022) respectively. Entries marked with '∗' report 95% confidence intervals.

The following linear and spectral models will be considered for evaluation: LINEAR ($\boldsymbol{XW}$), PROPAGATED LINEAR ($\boldsymbol{PXW}$), KERNEL ($\boldsymbol{P}^{(\mathcal{K})}\boldsymbol{XW}$), and LR KERNEL ($\boldsymbol{P}^{(r,\mathcal{K})}\boldsymbol{XW}$). Similar to the model hyperparameters, learning rate and weight decay of the optimizer, Adam (Kingma & Ba, 2015), will be determined using mean accuracies of the validation split of each dataset. For completeness, we have also implemented a non-linear baseline which learns using only feature information $\boldsymbol{X}$. This model will be a simple two-layer ReLU multi-layer perceptron MLP2 with hidden layer size determined through validation.

Our models and their results compared to other current SOTA methods can be found in Table 2. We note that, for most of the large graph benchmarks, our models perform within uncertainty or better compared to SOTA. In particular for directed graphs like Chameleon and Squirrel, we see gains in accuracy as high as 5% and 20% over other SOTA methods. A point of emphasis here is the relative simplicity of our models compared to the performance they attain. The absence of any post-model augmentations distinguishes our approach from other competing SOTA spectral methods (Wang & Zhang, 2022).

A point of difficulty where a performance gap persists, is where the node response $\boldsymbol{y}$ is overwhelmingly described by its nodal information $\boldsymbol{X}$. Graphs with this property (Actor, Cornell, Texas, and Wisconsin) can be identified by the negative performance gap between LINEAR and PROPAGATED LINEAR as well as the SOTA-like performance of MLP2. Note that, even without using any graph information, MLP2 is able to achieve SOTA within uncertainty on almost all of the $\boldsymbol{X}$-dominated, network datasets. Furthermore, in the cases of Cornell, Texas, and Wisconsin, we may be running into a sample size issues, as these dataset sare only 1/10 the size (a few hundred nodes) of the other benchmarks, and the networks are extremely sparse.

| Kernel | Cora | CiteSeer | PubMed | Chameleon | Squirrel | Actor | Cornell | Texas | Wisconsin |
|---|---|---|---|---|---|---|---|---|---|
| Identity | $78.8 \pm 2.7$ | $72.6 \pm 2.0$ | $81.6 \pm 0.9$ | $69.7 \pm 2.7$ | $44.9 \pm 2.9$ | $28.6 \pm 3.0$ | $60.4 \pm 8.1$ | $76.2 \pm 4.3$ | $71.6 \pm 5.7$ |
| Sob. Cmpct. | $75.1 \pm 1.9$ | $73.0 \pm 1.4$ | $88.5 \pm 0.4$ | $41.4 \pm 2.2$ | $33.2 \pm 1.1$ | $\mathbf{32.2 \pm 1.8}$ | $\mathbf{83.3 \pm 5.9}$ | $88.6 \pm 4.0$ | $\mathbf{92.1 \pm 3.4}$ |
| Linear | $81.1 \pm 2.0$ | $72.1 \pm 1.8$ | $82.3 \pm 1.0$ | $\mathbf{78.7 \pm 1.2}$ | $\mathbf{76.0 \pm 1.2}$ | $31.6 \pm 0.9$ | $66.5 \pm 6.1$ | $77.2 \pm 8.0$ | $81.3 \pm 4.8$ |
| Sob. Unbnd. | $88.8 \pm 0.8$ | $\mathbf{81.1 \pm 1.0}$ | $89.2 \pm 2.0$ | $54.5 \pm 6.4$ | $68.8 \pm 8.2$ | $30.7 \pm 1.0$ | $80.6 \pm 6.4$ | $\mathbf{88.2 \pm 2.6}$ | $90.4 \pm 5.6$ |
| Gauss. RBF | $\mathbf{88.6 \pm 1.0}$ | $80.3 \pm 1.9$ | $\mathbf{89.4 \pm 0.8}$ | $60.4 \pm 8.4$ | $71.3 \pm 4.4$ | $30.4 \pm 1.3$ | $79.4 \pm 5.3$ | $84.0 \pm 4.5$ | $85.8 \pm 4.7$ |

Table 3: Impact of the kernel choice on the performance of the full-rank KERNEL model. Bold entries correspond to the model selected by validation.

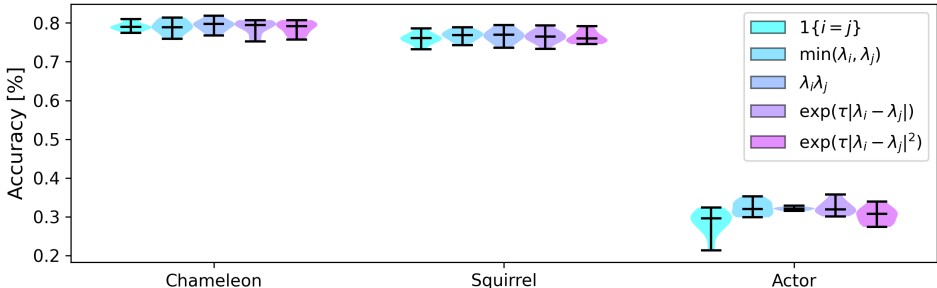

Figure 1: Performance homogenization achieved by LR KERNEL model on directed networks.

Future work should explore how to make these simple kernel methods, no worse than a linear model in the worst case. The introduction of an extra *regularization* parameter $\beta$ of the form

$$\boldsymbol{P}' = \boldsymbol{P} + \beta \boldsymbol{I} \tag{4}$$

may help here at the cost of minor complexity overhead. So far, preliminary implementations of equation 4 have not shown to be any more competitive than standard kernel approaches. It could be that a more complicated regularizing form is needed to balance the propagation and identity terms $\boldsymbol{P}$ and $\boldsymbol{I}$.

## 4 SPECTRAL KERNEL ABLATION

We next conduct an ablation study on the three degrees of freedom (kernel choice, matrix representation, and truncation factor) in constructing the propagation matrix $P$. Optimal choice of the kernel and other hyperparameters seem specific to particular datasets themselves. Although out-of-scope for the paper, one may consider contrasting the best and worst performing hyperparameters to gain insight into the underlying generative processes of these benchmark datasets.

For a first study, we consider ablating the kernel choice. Results of the ablation are shown in Table 3 for the full-rank KERNEL model, where a complicated dependence can be seen between the kernel choice and the accuracy of the estimated response. Although some results are within uncertainty, the dependence between kernel regularity and SSNC performance is not immediately clear. For Chameleon and Squirrel datasets, we see that the wrong kernel may lead to performance degradations up to $\sim 30\%$. This is a problem which is partially alleviated by the LR KERNEL model, where the option to reduce the kernel rank homogenizes some of the model performance. Figure 1 illustrates this homogenization effect. We stress however that this solution is partial, as the same order of homogenization is not observed for the undirected datasets. Identifying the relevant graph statistics which describe this homogenization discrepancy is something which is left to future work.

The next relevant hyperparameter is the matrix representation of the network. This choice can have an outsized impact on performance as the eigenvectors, otherwise known as the modes of the graph, are fixed once a representation is chosen. Similar to the kernel choice, it is not immediately clear when one matrix representation will outperform the other. Figure 2 shows the impact on performance is variable across both directed and undirected datasets.

Lastly, we carry out an ablation relative to the spectral truncation factor $r$. Larger spectral truncations have the benefit of accelerating model execution at the potential cost of performance. Figure 3

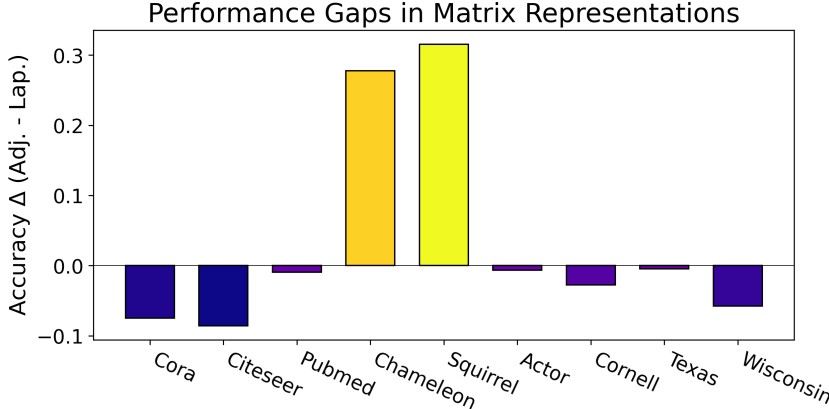

Figure 2: Accuracy comparison of the KERNEL model for different graph representations $A$ and $D - A$. Shown above is the signed accuracy difference between the adjacency and Laplacian representations. Best performing kernel was selected per dataset.

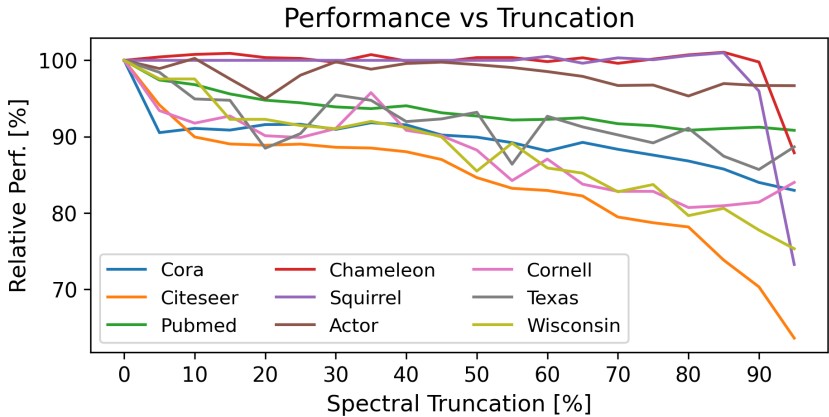

Figure 3: LR KERNEL performance relative to the full-rank KERNEL for different truncation factors $r$. Performance is seen to gradually decline on most datasets as the truncation factor $r$ increases. LR KERNEL performance can also be seen to periodically increase above full-rank KERNEL performance for the datasets Chameleon (red) and Squirrel (purple).

demonstrates how performance degrades gradually with the truncation factor. The rate at which performance degrades seems to be dependent on the dataset, but most benchmarks retain ∼90% performance even after a 50% spectral trunctation. In special cases like Squirrel and Chameleon, performance is even seen to increase at larger truncation values.

## 5 CHANGES IN EVALUATION CONVENTIONS

As interest in the SSNC learning task increased (Seeger, 2002; Zhu et al., 2003), so did the number of publicly available, real-world network datasets. These datasets spanned a variety of topics from citation networks Sen et al. (2008) to social co-occurences graphs Tang et al. (2009) to web link networks (Craven et al., 1998; Rozemberczki et al., 2021). From the modeling perspective, a common set of datasets was useful to benchmark different methods and one set of networks which quickly saw serialization were the citation datasets (Cora, Citeseer, Pubmed) popularized by Yang et al. (2016).

Yang et al. (2016) defined the "sparse" train-test split on the citation datasets and their node masks were made publically available. The sparse split had set 20 nodes per class for training and a 1000

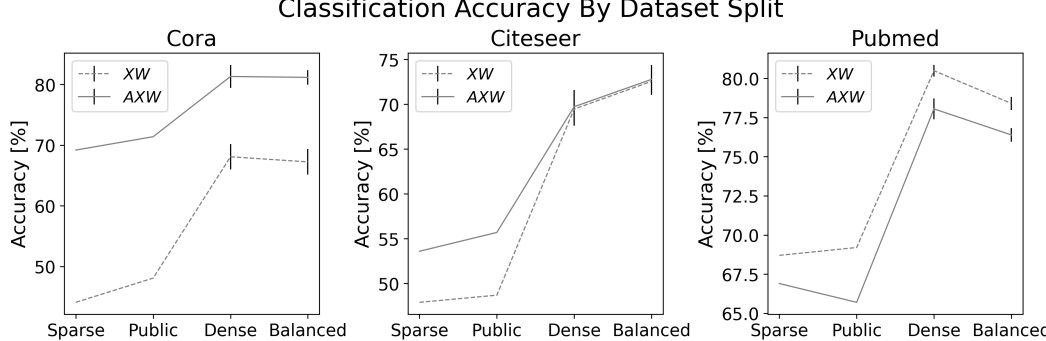

Figure 4: Accuracy results and uncertainties on the citation datasets using different splits with linear models $\boldsymbol{XW}$ and $\boldsymbol{AXW}$. "Public" refers to the split introduced by Kipf & Welling (2017). Both "Sparse" and "Public" are single splits, so one cannot associate uncertainty to them.

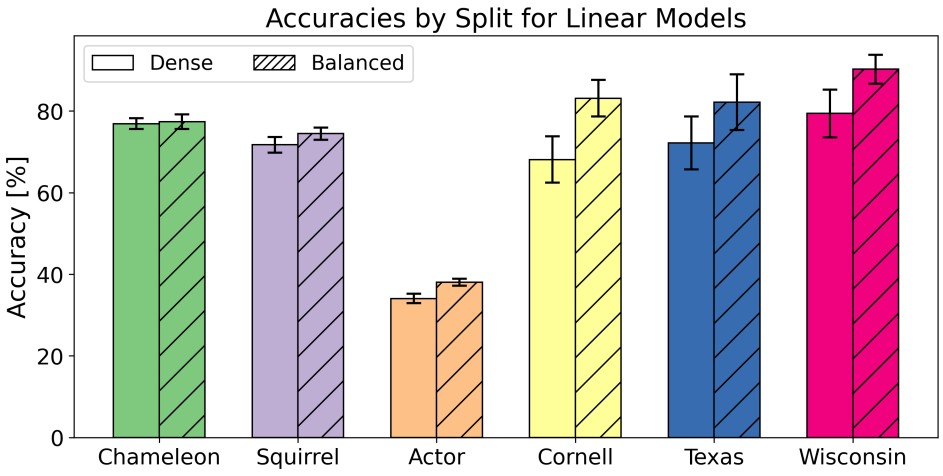

Figure 5: Accuracy results on datasets introduced by Pei et al. (2020). "Dense" refers to the original split while "Balanced" refers to the split introduced by Chien et al. (2021). Test results and uncertainties are evaluated using models $\boldsymbol{XW}$ and $\boldsymbol{AXW}$. Results shown are on the method with best validation per dataset.

nodes total for testing. These values were held constant for all three citation datasets, meaning larger networks like Pubmed were left with a total label rate of about $0.3\%$. Quickly following was the semi-supervised work of Kipf & Welling (2017) and Veličković et al. (2018). These follow-up papers considered an additional 500 previously unlabeled nodes to use as validation. In the respective code implementations of each paper, these additional labels were used in an early stopping criterion for the final model checkpoint.

Introduced later was the "dense" split by Pei et al. (2020), where train, validation, and test were now fractions of the whole graph, set to 60%-20%-20% respectively. This paper also popularized two new benchmark datasets, the WebKB dataset Craven et al. (1998) and the Wikipedia animal pages network Rozemberczki et al. (2021). After the introduction of these new benchmarks, a new "balanced" split was proposed by Chien et al. (2021). Here, each class in a network was masked according to a 60%-20%-20% split, which are then collected into the final train-validation-test splits. Both the new datasets and the balanced splits are common benchmarking practices for current SSNC papers and implementations of these conventions can be found in the code of various SOTA GNN papers (Luan et al., 2022; Wang & Zhang, 2022).

Provided in Figures 4-5 are visualizations on the impacts of different evaluation techniques on simple linear models ($\boldsymbol{XW}$ and $\boldsymbol{AXW}$). To keep things comparable to the sparse split, both the learning

rate ($10^{-3}$) and the weight decay (0.0) were fixed for the Adam optimizer. Despite this lack of tuning, note that the best of these models, per dataset, achieve roughly 85% and above of the performance relative to SOTA SSNC methods. For the high-end of this performance, see the classification results on the Squirrel dataset in Figure 5 where a mean accuracy of 77.3% is achieved.

New GNN architectures which make use of the more recent splitting techniques may also experience a performance bump similar to our linear models. This perhaps leads to an overstatement of the modeling contribution for certain new architecture and, on the downside, has the potential to persuade later researchers to incorporate unnecessary modeling complexities in their SSNC experiments. For this reason, we believe it is important to be upfront on the impact that the different splitting conventions have on performance.

## 6 CONCLUSIONS

We have shown how classically-inspired nonparametric techniques can be used to match, and sometimes exceed, previous spectral and non-linear GNN approaches. Our methods make no use of post-model augmentations such as dropout (Srivastava et al., 2014) or batchnorm (Ioffe & Szegedy, 2015) allowing for clean theoretical analysis in future work. We briefly note, that the formulation of the spectral kernel model itself may be of theoretical interest, as its simplified variants have ties to low-rank, noisy matrix sensing problems Fazel et al. (2008); Zhong et al. (2015); Deng et al. (2023).

Elaborating a little further, assume a regression setting with a scalar real-valued $y_i$, and let $\boldsymbol{\beta} \in \mathbb{R}^d$ take the place of our linear weights. In this case, our evaluation outputs will be scalar valued, so the $j$-th evaluation of the LR KERNEL model can be rearranged as

$$h_j(A, X) = \boldsymbol{e}_j^T \boldsymbol{U}(\mathrm{diag}(\boldsymbol{K\alpha})\boldsymbol{U}^T)\boldsymbol{X\beta}$$

$$= \sum_{i=1}^r \boldsymbol{e}_j^T \boldsymbol{u}_i (\boldsymbol{e}_i^T \boldsymbol{K\alpha}) \boldsymbol{u}_i^T \boldsymbol{X\beta}$$

$$= \sum_{i=1}^r (\widetilde{\boldsymbol{k}}_i^j)^T \boldsymbol{\alpha\beta}^T \widetilde{\boldsymbol{x}}_i$$

$$= \left\langle \sum_{i=1}^r \widetilde{\boldsymbol{k}}_i^j \widetilde{\boldsymbol{x}}_i^T, \boldsymbol{\alpha\beta}^T \right\rangle_F,$$

where $\widetilde{\boldsymbol{k}}_i^j = u_{i,j}\boldsymbol{K}\boldsymbol{e}_i$, $\widetilde{\boldsymbol{x}}_i = \boldsymbol{X}^T \boldsymbol{u}_i$, and $\langle \boldsymbol{A}, \boldsymbol{B} \rangle_F := \mathrm{tr}(\boldsymbol{A}^T \boldsymbol{B})$ is the Frobenius matrix inner product. This formulation has the goal to estimate a rank-1 matrix parameter $\boldsymbol{\alpha\beta}^T$ given $n$, rank-$r$ linear measurements of the form $\mathcal{A}_j(\cdot) = \langle \sum_{i=1}^r \widetilde{\boldsymbol{k}}_i^j \widetilde{\boldsymbol{x}}_i^T, \cdot \rangle_F$. If our underlying assumption is that adjacency $\boldsymbol{A}$ is noisy then the construction of $\widetilde{\boldsymbol{k}}_i^j$, and therefore our linear measurements, must be noisy as well.

On the empirical side, we explored pertinent hyperparameters to the spectral kernel model and showed how the dependence on these parameters may vary across different network datasets. On the low-rank side, we showed how spectral truncation can homogenize response outcomes for different kernel choices. Additionally, it was shown that performance declines gradually with increases in the truncation factor, pointing to practical speed-ups for non-parametric kernel implementations.

Lastly, we looked at how evaluation conventions on SSNC tasks have changed since the introduction of popular network datasets. We highlighted the recently defined balanced split and showed how its use can lead to increases in performance outside of what may be expected by uncertainty. By bringing attention to these changes, we hope to even the field on benchmark comparisons for later SSNC works, allowing future researchers to accurately compare their methods to previous SOTA results. The code for reproducing all the experiments will be made publicly available.

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
