# OpenReview forum: "Simplifying GNN Performance with Low Rank Kernel Models"
_ICLR.cc/2024/Conference — Submitted to ICLR 2024_

### Official Review · Reviewer_3FrS · 2023-10-28

**Soundness:** 2 fair
**Presentation:** 2 fair
**Contribution:** 2 fair
**Rating:** 3
**Confidence:** 4

**Summary:**

The authors posit that current GNN architectures may be over-engineered for semi-surpervised graph classification. Based on this posit, the authors propose a simplified one-layer GNN with a kernel. First, the adjacency matrix is applied with a low-rank singular value decomposition. Then, singular values are filtered by a kernel function. Finally, the kernel matrix is multiplied with a learnable parameter vector. By choosing different kernel function, the proposed GNN achieves great performance in semi-surpervised graph classification.

**Strengths:**

1. The idea of employing kernel functions to filter eigenvalues or singular values of the adjacency matrix is intriguing.
2. This paper demonstrates that the recent advancements in GNN performance may be partially ascribed to shifts in evaluation conventions. This revelation holds significance for follow-up research on GNNs.

**Weaknesses:**

1. The overall time complexity of the proposed method is considerably high, particularly during the low-rank singular value decomposition of the adjacency matrix.
2. The rationale behind selecting identity, linear, compact Sobolev, unbounded Sobolev, and Gaussian radial basis function kernels remains unexplained.

**Questions:**

1. Figure 3 illustrates a compelling result that appears to contradict the intuitive notion that the expressive capability of the kernel enhances as the rank increases. Can the authors elucidate the underlying factors contributing to this unexpected experimental result?

---

### Official Review · Reviewer_3E1j · 2023-10-30

**Soundness:** 2 fair
**Presentation:** 2 fair
**Contribution:** 2 fair
**Rating:** 3
**Confidence:** 3

**Summary:**

This paper revisits spectral GNN to semi-supervised node classification (SSNC) and conduct some experiments to show that simple spectral GNN techniques can already lead to good performance for SSNC. The presented results are reasonable and solid.

**Strengths:**

The reviewer would agree on the main purpose of this work, i.e.,  simple spectral approaches may be sufficient to achieve SOTA or comparable performance for SSNC, compared to many existing complicated methods.

**Weaknesses:**

However, the reviewer finds it's a bit difficult to catch up the novelty or new insight revealed by this work, and there are some parts in the paper which should definitely clarified or even re-organized.

**Questions:**

### Major:
1. In sec. 1, the authors mention "we simultaneously simplify and generalize previous spectral approaches to SSNC while maintaining or exceeding previous performance benchmarks." This seems the technique examed by this work. However, the details of simpliciation techniques and the pipeline are lacking, as the first paragraphs in sec. 3 are enough and systematic.
2. Applying nonlinearity to the spectrum is not a new thing. Then, the difference or the particular interest of eveluations leveraging such technique seems unclear in this work.
3. Despite of the paragraph under Eq.(2), the procedures and complexity of how the $K$ and $K\alpha$ are computed are still unclear. Should we compute a kernel matrix $K$ first and compute its eigenvectors $\alpha$? If so, it would take some computation especially in an iterative update scheme.
4. In the paragraph **spectral truncation factor**, do we need to compute the SVD of $A$? What if the graph is really big?
5. How is the hyperparamter of the SBF kernel tuned?
6. In sec. 2, the authors mentioned "consider a one layer linear GNN", so is it consistent to the experiment in Sec. 3? If so, how about the multiple-layer case evaluations? As mentioned in bulltet 1, the reviewer would expect a clearer and more symtematic view on how the GNN is built and how the experiments are proceeded under which setups.
7. The evaluated graphs are not large enough. The reviewer would wonder how the authors think or what the evaluation results would be for large-scale graphs. Does the low-rank simplification property still hold competence with complicated methods?


### Minor:
1. Please unify the notations of vectors, matrices, etc.
2. For directed networs, the number of evaluated datasets is a bit too less.
3. The results in Fig. 3 seem a common sense, as the less being truncated the more information being reserved. For different datasets, the reasonable percents of truncation vary a lot, which is reasonable for the tasks, but what would be the particular interest revealed in fig. 3 for this work? It seems unclear. By changing the kernels, the reviewer would expect different results (despite similar trend for some datasets).
4. What would be the difference with the kernel spectral method, or is the kernel spectral method also applicable? In kernel specral method, the data matrix, e.g., the adjacency matrix $A$ is applied with weighted KPCA problem and conducts a (generalized) eigenvalue problem on the kernel matrix [1], or the adjacency matrix $A$ is applied with the asymmetrick kernel SVD techniques for performing the singular value decomposition.
5. Either in sec.3 or somewhere, the reviewer would suggest to have an overall sketch on the roles and main purposes for each set of experiments in sec.3-5, which shall improve readability and present the readers from checking paragraphs and sections back and forth.
6. For most of the cases evaluated, the performance gaps are still substantial.
7. Can the efficiency be compared?


[1] Alzate C., Suykens J. A. K., Multiway Spectral Clustering with Out-of-Sample Extensions through Weighted Kernel PCA, IEEE Transactions on Pattern Analysis and Machine Intelligence, 2010.

[2] Tao Q., Tonin F., Patrinos P., Suykens J. A.K., Nonlinear SVD with Asymmetric Kernels: feature learning and asymmetric Nystrom method, 2023

---

### Official Review · Reviewer_g7wG · 2023-10-31

**Soundness:** 3 good
**Presentation:** 3 good
**Contribution:** 2 fair
**Rating:** 5
**Confidence:** 3

**Summary:**

This paper discusses the use of simpler, traditional methods from nonparametric estimation to improve the performance of Graph Neural Networks (GNNs) in semi-supervised node classification (SSNC). The authors argue that many current GNN architectures are over-engineered and that conventional techniques applied in the spectral domain can achieve state-of-the-art performance on SSNC benchmarks. They also examine recent performance improvements in GNN approaches and attribute them partly to shifts in evaluation conventions.

**Strengths:**

Originality: The paper presents a novel perspective on Graph Neural Networks (GNNs) by arguing that many current GNN architectures are over-engineered and that traditional, simpler methods can achieve state-of-the-art performance on semi-supervised node classification (SSNC) benchmarks. The authors suggest using conventional techniques from nonparametric estimation in the spectral domain to improve GNN performance. *This original approach challenges the prevailing notion that complex GNN architectures are necessary for high performance.*

Quality: The paper provides a thorough ablative study on the hyperparameters associated with GNN spectral filtering techniques. It investigates the impact of different propagation structures, spectral truncation factors, and kernel choices. The *experimental results demonstrate that the proposed simpler GNN models perform on par with or better than current state-of-the-art methods on benchmark datasets for SSNC*. The study reflects a high quality of research methodology and experimental rigor.

Clarity: The paper is well-written and effectively presents its arguments, findings, and methodologies. The document snippets provide concise summaries that cover various aspects of the paper, such as the experiments conducted, evaluation conventions, and evaluation results. The organization of the document snippets allows for a clear understanding of the different sections and their content.

Significance: The paper’s significance lies in its ability to simplify modeling considerations for GNNs while achieving state-of-the-art or near-state-of-the-art results. By highlighting the importance of spectral reshaping and modeling techniques, the paper suggests that GNN architectures can be streamlined without sacrificing performance. This has implications for the development of more interpretable GNN models and heuristics for SSNC problems. Additionally, the paper addresses recent shifts in evaluation conventions and provides insights into improving performance on SSNC benchmarks.

**Weaknesses:**

**Insufficient Analysis on the Experimental Results**

While the paper presents an ablative study on the hyperparameters associated with GNN spectral filtering techniques, the analysis is limited to a specific set of experiments. It would greatly benefit the work if the authors conducted a deeper analysis for the empirical findings. This would help in understanding the robustness and limitations of the proposed models.


**Lack of Code for Reproductivity**

The reliance on experimental results in this work is evident, but the statement that "The code for reproducing all the experiments will be made publicly available" is considered unacceptable.

**Questions:**

**Q1.**  How can we theoretically analyze the experimental results and understand the effectiveness of such a straightforward spectral method? Despite the valuable insights provided by the experimental results, it seems that the paper may lack a sufficient level of theoretical insight. A more in-depth analysis of the empirical findings would greatly enhance the work and contribute to a better understanding of the robustness and limitations of the proposed models.

**Q2.**  The paper states that the proposed simpler GNN models perform comparably to or better than current state-of-the-art methods on benchmark datasets for SSNC. Can the authors discuss the potential limitations of these simpler models and any trade-offs in terms of interpretability and model complexity?

**Q3.**  In the conclusion, the authors suggest the potential for more interpretable models and heuristics for future SSNC problems. Can the authors provide some insights into how their proposed simpler models contribute to interpretability and suggest directions for future research in this area?

---

### Official Review · Reviewer_3XZw · 2023-10-31

**Soundness:** 2 fair
**Presentation:** 2 fair
**Contribution:** 1 poor
**Rating:** 1
**Confidence:** 4

**Summary:**

This paper revisits the application of spectral GNN approaches to semi-supervised node classification (SSNC) and raises the question of whether the current complex GNN architectures are necessary. The authors argue that traditional methods from nonparametric estimation applied in the spectral domain might replace deep learning-inspired GNN designs. They claim that these conventional techniques can achieve state-of-the-art performance on common SSNC benchmarks. They also argue that some of the recent performance improvements of complex methods might be attributed to shifts in evaluation conventions. However, I have several concerns about the novelty of this paper.

**Strengths:**

Addressing the use of simpler methods to match and outperform complex DNN-based architectures.

**Weaknesses:**

Lack of Novelty: The paper does not present significant novelty. While it challenges the use of complex GNNs with spectral approaches, it falls short of providing a strong theoretical analysis or thorough experimental validation to support its claims. To enhance the paper's contribution, it should either include a rigorous theoretical analysis or conduct additional comprehensive experiments, preferably in supplementary material.

Limited Experiments: The paper contains a limited number of experiments, which is insufficient given the broad claims made. A supplementary material section should be added to provide a more thorough evaluation, including comparisons with existing methods, diverse datasets, and various hyperparameter settings.

**Questions:**

- What motivated your choice of the spectral truncation factor and the choice of kernel functions? Did you conduct any sensitivity analysis on these choices?

- Have you considered comparing your proposed approach with existing GNN architectures on a broader set of benchmark datasets to ensure the generality of your claims? The paper lacks an extensive experimental investigation on multiple benchmarks and algorithms with several ablation studies regarding the claim of the paper? Or rather a theoretical study of the phenomenon observed.

- Could you elaborate on the limitations and assumptions of your method? Are there scenarios where your approach may not be as effective, and if so, what are they?

- There are multiple typos in the paper. Could you please proofread?

---

### Meta-Review · Area_Chair_qTDS · 2023-12-09

**Metareview:**

The paper studies semi-supervised node classification — a problem in which we have access to a graph, with adjacency matrix A, features X for each vertex of the graph, and labels for a subset of the vertices. The goal is to predict the remaining labels. A variety of graph neural networks have been proposed for this task. Graph neural networks repeatedly apply (i) a feature transformation, (ii) a propagation operation along the graph, and a nonlinearity; in popular networks, these operations can be learned on a per-layer basis. The paper argues that some of this complexity may be unnecessary for achieving near state-of-the-art results on popular SSNC benchmarks. It studies alternative linear predictors, which involve a linear mapping W on features and a linear propagation operator P across nodes. The propagation operator is generated from the spectral structure of the adjacency matrix or Laplacian, by applying a kernel mapping to the eigenvalues. The paper observes that this relatively simple network structure achieves near state-of-the-art performance on a number SSNC datasets. The paper also studies the effect of various splitting conventions on performance, arguing that the ``balanced split’’ which is used in many recent GNN works itself has a strong positive effect on performance, even for simple networks.

Overall, the main contribution of the paper is to question the need for complex GNN architectures, by showing that simpler architectures can achieve comparable performance, and to raise issues around experiment design. Another technical novelty is the paper’s spectral kernel propagation operator.

Reviewers expressed appreciation for the paper’s effort to simplify GNN structures, while raising concerns about the motivation and explanation for kernel spectral operations, the extensiveness and reproducibility of the experiments, and the time complexity of spectral operations on the adjacency matrix, evaluating the paper as below the threshold for acceptance.

**Justification For Why Not Higher Score:**

Reviewers raised a number of concerns, including the motivation for particular spectral kernels, the computational cost of these operations for large adjacency matrices, and the extensiveness of the experiments.

**Justification For Why Not Lower Score:**

N/A

---

### Decision · Program_Chairs · 2024-01-16

Reject